# Physics-Informed Diffusion Models in Spectral Space

**Davide Gallon** [* 1 2]   **Philippe von Wurstemberger** [* 3]   **Patrick Cheridito** [1]   **Arnulf Jentzen** [4 2]

## Abstract

We propose *physics-informed spectral diffusion* (PISD), a methodology that combines generative latent diffusion models with physics-informed machine learning to generate solutions of *partial differential equations* (PDEs) conditioned on partial observations, which includes, in particular, forward and inverse PDE problems. We learn the joint distribution of PDE parameters and solutions via a diffusion process in a latent space of scaled spectral representations, where Gaussian noise corresponds to functions with controlled regularity. This spectral formulation enables significant dimensionality reduction compared to grid-based diffusion models and ensures that the induced process in function space remains within a class of functions for which the PDE operators are well defined. Building on diffusion posterior sampling, we enforce physics-informed constraints and measurement conditions during inference, applying Adam-based updates at each diffusion step. We evaluate the proposed approach on Poisson, Helmholtz, and incompressible Navier–Stokes equations, demonstrating improved accuracy and computational efficiency compared with existing diffusion-based PDE solvers, which are state of the art for sparse observations. Code is available at https://github.com/deeplearningmethods/PISD.

## 1. Introduction

Deep learning approaches for PDEs have progressed toward increasingly general representations of solution spaces:

---
[*]Equal contribution  [1]Department of Mathematics, ETH Zurich, Switzerland [2]Institute for Analysis and Numerics, University of Münster, Germany [3]School of Data Science, The Chinese University of Hong Kong, Shenzhen (CUHK-Shenzhen), China [4]School of Data Science and School of Artificial Intelligence, The Chinese University of Hong Kong, Shenzhen (CUHK-Shenzhen), China. Correspondence to: Davide Gallon <davide.gallon@uni-muenster.de>.

*Proceedings of the 43rd International Conference on Machine Learning*, Seoul, South Korea. PMLR 306, 2026. Copyright 2026 by the author(s).

from *physics-informed neural networks* (PINNs) (Raissi et al., 2019; Sirignano & Spiliopoulos, 2018), which approximate individual PDE instances via residual-based objectives, to neural operators (Anandkumar et al., 2019; Kovachki et al., 2023; Lu et al., 2021; Li et al., 2021; 2024b), which learn solution maps for families of parametric PDEs, and more recently to generative models that learn distributions over PDE parameters and solutions (Huang et al., 2024; Ciftci & Hackl, 2024). Through conditional sampling, the generative perspective naturally supports fundamentally different problem settings: *forward problems*, inferring the PDE solution from sparse or full observations of the coefficient; *inverse problems*, inferring the PDE coefficient from sparse or full observations of the solution; and *joint reconstruction*, recovering both the solution and coefficient from sparse observations of the solution and coefficient. Many of these problems, especially those with sparse observations, are ill-posed in the classical sense and lie outside the reach of standard solvers. The generative approach addresses this through a Bayesian formulation, in which the learned prior regularizes the problem and induces a well-defined posterior.

In this work, we develop such a generative framework, which we term *physics-informed spectral diffusion* (PISD). Our approach represents functions via scaled spectral coefficients and trains a diffusion model (Ho et al., 2020; Song et al., 2021; Karras et al., 2022) in the resulting finite-dimensional latent space (Rombach et al., 2022; Vahdat et al., 2021). The scaling is obtained from the data distribution and ensures that the diffusion process induced in function space remains within a class of functions for which the underlying PDE operators are well defined. In particular, we show that if the data distribution satisfies a Sobolev regularity condition, then the induced diffusion process in function space preserves this regularity. At inference time, we enforce physics-informed constraints and measurement conditions using a variant of *diffusion posterior sampling* (DPS) (Chung et al., 2023a) with Adam-based updates (Kingma & Ba, 2014). We present a schematic of the PISD method in Figure 1 and describe it in detail in Section 3.

**Contributions.**

1. We introduce PISD, a generative framework that learns joint distributions of PDE parameters and solutions

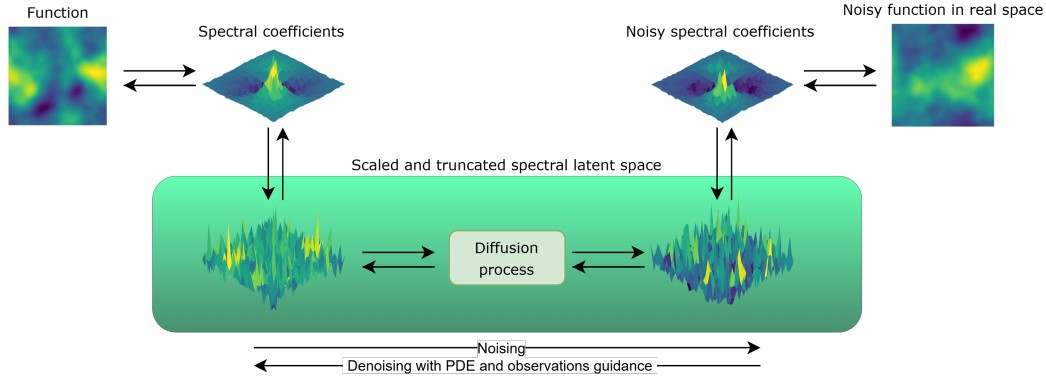

*Figure 1.* Overview of physics-informed spectral diffusion (PISD). The model learns to generate function valued solutions by performing a diffusion process in a latent space of scaled spectral function representations. At inference, PDE and observation constraints are enforced via Adam-based guidance.

using diffusion models in a scaled spectral latent space with physics-informed guidance at inference.

2. We establish that appropriate scaling of spectral coefficients induces a diffusion process in function space with controlled Sobolev regularity, ensuring that PDE operators are well defined throughout the generative process.

3. We demonstrate that the spectral formulation enables substantial dimensionality reduction compared to grid-based diffusion models, leading to significant computational speedups.

4. We extend DPS with Adam-based inference-time updates and show improvements over standard *gradient descent* (GD)-based guidance.

We test our method on Poisson, Helmholtz, and incompressible Navier–Stokes equations. In our experiments, the PISD method reduces inference time by a factor of 3 to 15 relative to state-of-the-art baselines, while matching or improving accuracy by up to a factor of 10.

## 2. Related Work

**Physics-informed diffusion models.** Several recent works combine diffusion models with physics-informed constraints. DiffusionPDE (Huang et al., 2024) and CoCo-Gen (Jacobsen et al., 2025) enforce PDE constraints via residual gradients during sampling, but operate on grid-based discretizations with finite differences. In the infinite-resolution limit, the PDE residual becomes ill-defined since the Gaussian noise considered in the diffusion process converges to spatial white noise. Consequently, these methods apply PDE guidance only near the end of the reverse process when the function is already somewhat regular. We compare against DiffusionPDE in Section 4 and show that

PISD achieves significantly lower PDE residuals. Physics-informed diffusion models in Bastek et al. (2024) similarly use grid-based finite differences but enforce PDE constraints during training rather than inference. Physics-informed diffusion models in Shu et al. (2023) address flow field reconstruction with a method closely related to DiffusionPDE. Pi-Fusion (Qiu et al., 2024) also uses a physics-informed guidance term during inference, but only considers forward problems. FunDiff (Wang et al., 2025) operates in a learned latent space with a continuous vision transformer decoder that can be differentiated at arbitrary locations. However, PDE constraints are enforced only during encoder training, not at inference.

**Diffusion models for PDEs.** Other approaches use diffusion models to generate PDE solutions without physics-informed losses. FunDPS (Yao et al., 2025; Mammadov et al., 2024) extends DPS to function-space-valued diffusion processes, providing a rigorous theoretical foundation for the solution of inverse problems with diffusion models in function spaces. Several methods incorporate neural operators into the denoiser architecture (Oommen et al., 2024; Hu et al., 2024; Yang & Sommer, 2023). Wavelet diffusion neural operator (Hu et al., 2024) considers diffusions in the space of wavelet coefficients and is thus related to our spectral approach but targets functions with abrupt changes rather than ensuring smoothness. Additional diffusion-based PDE methods that do not employ physics-informed losses or function-space formulations include Li et al. (2024a); Shysheya et al. (2024).

**Diffusion models in infinite-dimensional function spaces.** A growing body of work extends diffusion models to infinite-dimensional function spaces. Spectral diffusion processes (Phillips et al., 2022) formulate diffusion in the space of spectral coefficients, similar to our approach, however without the data-dependent scaling that ensures regularity

throughout the diffusion process. Kerrigan et al. (2023) generalize discrete-time diffusion models (Ho et al., 2020) to infinite-dimensional function spaces, explicitly considering diffusion processes over functions with prescribed Sobolev regularity. Pidstrigach et al. (2024); Hagemann et al. (2023); Lim et al. (2023); Franzese et al. (2023); Lim et al. (2025) develop continuous-time diffusion models in infinite-dimensional function spaces, with particular attention to consistency across discretization levels. Na et al. (2025) generalizes the probability-flow *ordinary differential equation* (ODE) to infinite-dimensional function spaces.

**Guidance with advanced optimizers.**  Several works improve the GD-based guidance in DPS by employing more advanced optimizers. Concurrently and independently, Belardi et al. (2026) propose the same Adam-based replacement as we do and validate it extensively on image generation tasks. Chung et al. (2023b) incorporate *alternating direction method of multipliers* (ADMM) into the diffusion inference process to enforce data-consistency constraints. Wang et al. (2024) unroll the entire reverse diffusion process and apply Adam updates to the final output rather than during each sampling step. Xu et al. (2025) perform multiple GD and projection steps per diffusion step, motivated by the observation that DPS guidance aligns more closely with maximum a posteriori estimation than with conditional score estimation.

## 3. Method

### 3.1. Problem setting

We consider an abstract PDE problem given by a PDE residual functional $\mathcal{L}_{\mathrm{PDE}} \colon \mathcal{F} \to [0, \infty)$ defined on a Hilbert space $\mathcal{F}$. Our goal is to generate samples $f \in \mathcal{F}$ satisfying $\mathcal{L}_{\mathrm{PDE}}(f) = 0$, typically conditioned on additional constraints such as boundary conditions or sparse measurements. To this end, we assume a prior distribution $\nu \colon \mathcal{B}(\mathcal{F}) \to [0, 1]$ supported on PDE solutions, meaning that

$$\nu(\{f \in \mathcal{F} \colon \mathcal{L}_{\mathrm{PDE}}(f) = 0\}) = 1.$$

In addition, we denote by $X_{\mathrm{data}} \in \mathcal{F}$ a random variable which is distributed according to $\nu$. In practice, we assume that we can draw samples of $X_{\mathrm{data}}$ by computing approximate PDE solutions with a classical solver, such as a finite difference or finite element method.

*Example* 3.1 (Poisson equation). To make the above setting concrete, we specify it for a 2-dimensional Poisson equation. Let $\Omega = (0, 1)^2$, $\mathcal{A} = L^2(\Omega)$, let $\mathcal{S} = H^2(\Omega) \cap H_0^1(\Omega)$ be the Sobolev space of twice weakly differentiable functions which vanish at the boundary, let $O \colon \mathcal{A} \to \mathcal{S}$ be the solution operator which assigns to all $a \in \mathcal{A}$ the weak solution $u \in \mathcal{S}$

of the Poisson equation

$$\Delta u = a \quad \text{on} \quad \Omega$$

with zero boundary conditions, and let $A_{\mathrm{data}} \in \mathcal{A}$ be a random variable (see, e.g., Evans (2010, Chapters 5 and 6) for definitions of Sobolev spaces and well-posedness of the considered PDE problem). We then choose $\mathcal{F} = \mathcal{S} \times \mathcal{A}$, $X_{\mathrm{data}} = (O(A_{\mathrm{data}}), A_{\mathrm{data}})$ and define for all $(u, a) \in \mathcal{F}$ that

$$\mathcal{L}_{\mathrm{PDE}}((u, a)) = \|\Delta u - a\|_{L^2}^2.$$

### 3.2. Diffusion model in spectral space

To approximately generate function samples from $\nu$, we suggest to use a diffusion model over a finite-dimensional spectral encoding $\mathcal{E} \colon \mathcal{F} \to \mathbb{R}^l$ of functions in $\mathcal{F}$. In all our experiments, the function $\mathcal{E}$ will correspond to a suitably truncated and normalized Fourier transform (cf. Section 3.4 below). We denote by $\mathcal{I} \colon \mathbb{R}^l \to \mathcal{F}$ the approximate inverse transform of $\mathcal{E}$ for which we have for all $f \in \mathcal{F}$ that

$$\mathcal{I}(\mathcal{E}(f)) \approx f. \tag{1}$$

Following Karras et al. (2022), we train a denoiser $D \colon \mathbb{R}^p \times \mathbb{R}^l \times (0, \sigma_{\max}] \to \mathbb{R}^l$ with $p \in \mathbb{N}$ trainable parameters to approximate for all noise levels $\sigma \in (0, \sigma_{\max}]$ that

$$D_{\theta^*}(\hat{X}_{\mathrm{data}} + \sigma N, \sigma) \approx \hat{X}_{\mathrm{data}}, \tag{2}$$

where $\hat{X}_{\mathrm{data}} = \mathcal{E}(X_{\mathrm{data}})$ is the spectral encoding of the data $X_{\mathrm{data}}$ and $N \sim \mathcal{N}(0, I_l)$ is a Gaussian random variable independent of $X_{\mathrm{data}}$. For a suitable noise schedule $\sigma \colon [0, T] \to (0, \sigma_{\max}]$, we then expect reverse-time solutions of the ODE

$$\mathrm{d}\hat{x}_t = -\dot{\sigma}_t(\sigma_t)^{-1}(D_{\theta^*}(\hat{x}_t, \sigma_t) - \hat{x}_t) \, \mathrm{d}t \tag{3}$$

for $t \in [0, T]$ with $\hat{x}_T \sim \mathcal{N}(0, \sigma_{\max}^2 I_l)$ to gradually remove noise so that the initial value $\hat{x}_0 \overset{d}{\approx} \hat{X}_{\mathrm{data}}$ is approximately distributed like the spectral encoding of the data $\hat{X}_{\mathrm{data}}$. With (1) it thus follows that $\mathcal{I}(\hat{x}_0)$ is approximately distributed according to $\nu$.

### 3.3. Physics-informed and measurement guidance

In the next step, we add two guidance terms to the ODE in (3) to, first, help the model enforce the PDE conditions, and, second, generate samples conditioned on partial measurements. More specifically, for a measurement operator $\mathcal{M} \colon \mathcal{F} \to \mathbb{R}^m$ and a given measurement $y \in \mathbb{R}^m$, we want to ensure that

$$\mathcal{M}(\mathcal{I}(\hat{x}_0)) = y \qquad \text{and} \qquad \mathcal{L}_{\mathrm{PDE}}(\mathcal{I}(\hat{x}_0)) = 0. \tag{4}$$

For this we use the DPS technique developed in Chung et al. (2023a); Huang et al. (2024) which enables weak enforcement of guidance conditions in diffusion models by adding

forcing terms to the backward diffusion process in the direction of the gradient of the target quantities. Applying this to the reverse-time ODE in (3) with the conditions in (4), and introducing guidance weights $\lambda_{\text{obs}}, \lambda_{\text{PDE}} \in [0, \infty)$, we obtain the guided reverse-time ODE:

$$
\begin{aligned}
\mathrm{d}\hat{x}_t = \Bigg( & -\dot{\sigma}_t(\sigma_t)^{-1}(D_{\theta^*}(\hat{x}_t, \sigma_t) - \hat{x}_t) \\
& + \lambda_{\text{obs}} \nabla_{\hat{x}_t} \Big[ \|y - \mathcal{M}(\mathcal{I}(D_{\theta^*}(\hat{x}_t, \sigma_t)))\|^2 \Big] \\
& + \lambda_{\text{PDE}} \nabla_{\hat{x}_t} \Big[ \mathcal{L}_{\text{PDE}}(\mathcal{I}(D_{\theta^*}(\hat{x}_t, \sigma_t))) \Big] \Bigg) \mathrm{d}t \quad (5)
\end{aligned}
$$

for $t \in [0, T]$ with $\hat{x}_T \sim \mathcal{N}(0, \sigma_{\max}^2 I_l)$. To obtain a concrete algorithm from (5), it remains to discretize the ODE in reverse time. We observe that, under a reverse-time Euler discretization of the ODE in (5), the contributions of the guidance terms correspond to GD steps. Motivated by this, we suggest to replace these GD updates by a more advanced gradient-based optimizer such as Adam (Kingma & Ba, 2014). We show empirically that this leads to significantly better results than standard GD (see Appendix C). We present the resulting PISD method in Algorithm 1.

---

**Algorithm 1** PISD

---

**Training Phase**

**Require:** PDE loss functional $\mathcal{L}_{\text{PDE}} \colon \mathcal{F} \to [0, \infty)$, data set $X_1, \ldots, X_M \in \mathcal{F}$ such that $\mathcal{L}_{\text{PDE}}(X_i) = 0$ for all $i \in \{1, \ldots, M\}$

1: Choose $\mathcal{E} \colon \mathcal{F} \to \mathbb{R}^l$, $\mathcal{I} \colon \mathbb{R}^l \to \mathcal{F}$ based on $X_1, \ldots, X_M$ (cf. Section 3.4)
2: Choose denoiser $D \colon \mathbb{R}^p \times \mathbb{R}^l \times (0, \sigma_{\max}] \to \mathbb{R}^l$
3: $\theta^* \leftarrow \underset{\theta \in \mathbb{R}^p}{\operatorname{argmin}} \mathbb{E}\Big[ \|D_\theta(\mathcal{E}(X_i) + \sigma N, \sigma) - \mathcal{E}(X_i)\|^2 \Big]$
   with $(i, \sigma, N) \sim \mathcal{U}_{\{1,\ldots,M\}} \times \mathcal{U}_{[0,\sigma_{\max}]} \times \mathcal{N}(0, I_l)$

---

**Inference/Sampling Phase**

**Require:** Number of steps $N \in \mathbb{N}$, noise schedule $(\sigma_n)_{n \in \{0,\ldots,N\}} \subseteq (0, \sigma_{\max}]$, measurement operator $\mathcal{M} \colon \mathcal{F} \to \mathbb{R}^m$, measurement $y \in \mathbb{R}^m$, guidance weights $\lambda_{\text{obs}}, \lambda_{\text{PDE}} \in [0, \infty)$

1: ADAM $\leftarrow$ initialize Adam optimizer on $\mathbb{R}^l$
2: $\hat{x} \sim \mathcal{N}(0, \sigma_{\max}^2 I_l)$
3: **for** $n = 1$ to $N$ **do**
4: $\quad G_{\text{obs}} \leftarrow \nabla_{\hat{x}} \Big[ \|y - \mathcal{M}(\mathcal{I}(D_{\theta^*}(\hat{x}, \sigma_n)))\|^2 \Big]$
5: $\quad G_{\text{pde}} \leftarrow \nabla_{\hat{x}} \Big[ \mathcal{L}_{\text{PDE}}(\mathcal{I}(D_{\theta^*}(\hat{x}, \sigma_n))) \Big]$
6: $\quad \hat{x} \leftarrow \hat{x} - (\sigma_n)^{-1}(D_{\theta^*}(\hat{x}, \sigma_n) - \hat{x})(\sigma_{n-1} - \sigma_n)$
7: $\quad \hat{x} \leftarrow \hat{x} - \text{ADAM}(\lambda_{\text{obs}} G_{\text{obs}} + \lambda_{\text{PDE}} G_{\text{pde}})$
8: **end for**
9: **Return** $\mathcal{I}(\hat{x})$

---

### 3.4. Spectral encoding

In this section we specify the encoding $\mathcal{E}$ introduced in Section 3.2. Our approach is based on truncated spectral representations of functions in $\mathcal{F}$, combined with a frequency-wise normalization determined from the data distribution. Spectral representations are natural in the PDE context, as they allow for explicit evaluation of differential operators.

The normalization plays a central role in our method and is therefore treated as part of the encoding rather than as a standard preprocessing step. Unlike conventional data normalization, our scaling is applied in latent space rather than in physical space, is performed independently for each spectral coefficient, and, because the latent variables correspond to Fourier modes, induces a form of regularization in function space. As we show below, this scaling ensures that Gaussian noise in the latent space corresponds to functions with controlled regularity, which is essential to ensure that PDE operators are well defined throughout the diffusion process.

For concreteness, we present an encoding for the case $\mathcal{F} = L^2(\mathbb{T}^d)$, where $\mathbb{T} = \mathbb{R}/\mathbb{Z}$ denotes the torus. When $\mathcal{F}$ consists of tuples of functions, as in Example 3.1, we apply our encoding approach component-wise and concatenate the results. We consider complex-valued functions and Fourier series here for ease of presentation; in our experiments we sometimes also work with real-valued functions using sine or cosine series.

Let $(\phi_n)_{n \in \mathbb{Z}^d} \subseteq L^2(\mathbb{T}^d)$ be the Fourier basis on $\mathbb{T}^d$ given by

$$
\forall n \in \mathbb{Z}^d \colon \quad \phi_n(x) = e^{2\pi i \langle n, x \rangle} \quad (6)
$$

and for all $f \in L^2(\mathbb{T}^d)$, $n \in \mathbb{Z}^d$ let $\hat{f}(n) = \langle f, \phi_n \rangle_{L^2(\mathbb{T}^d)}$ denote the $n$-th Fourier coefficient of $f$. To normalize the spectral representation, we define for each $n \in \mathbb{Z}^d$ the standard deviation

$$
(s_n)^2 = \text{Var}\big(\widehat{X_{\text{data}}}(n)\big). \quad (7)
$$

Given a truncation set $K \subseteq \mathbb{Z}^d$ with $|K| = l$, we define the encoding $\mathcal{E} \colon \mathcal{F} \to \mathbb{C}^K \cong \mathbb{C}^l$ and its inverse $\mathcal{I} \colon \mathbb{C}^K \to \mathcal{F}$ for all $f \in \mathcal{F}$, $n \in K$, $\alpha \in \mathbb{C}^K$ by

$$
\mathcal{E}(f)(n) = \frac{\hat{f}(n)}{s_n} \quad \text{and} \quad \mathcal{I}(\alpha) = \sum_{n \in K} s_n \alpha(n) \phi_n. \quad (8)
$$

The following lemma shows that this scaling ensures that Gaussian noise in the latent space corresponds to a suitably regular function in $\mathcal{F}$, provided the data exhibits the same regularity.

**Lemma 3.2.** *Let $k \in \mathbb{N}$, assume $\mathbb{E}\big[\|X_{\text{data}}\|_{H^k(\mathbb{T}^d)}^2\big] < \infty$, let $(Z_n)_{n \in \mathbb{Z}^d}$ be i.i.d. $\mathcal{N}(0,1)$ random variables, and let $N \in L^2(\mathbb{T}^d)$ be the random function given by*

$$
N = \sum_{n \in \mathbb{Z}^d} s_n Z_n \phi_n. \quad (9)
$$

*Then $\mathbb{E}\big[\|N\|_{H^k(\mathbb{T}^d)}^2\big] < \infty$.*

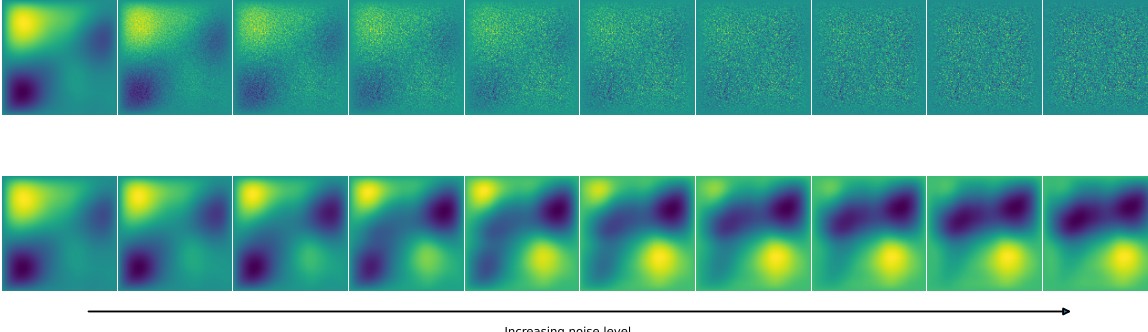

*Figure 2.* Comparison of the forward processes in function space: The top row shows heatmaps of the function values resulting from a standard grid-based model. The function becomes increasingly noisy and irregular. In contrast, the bottom row illustrates PISD where the function maintains a smooth, regular structure throughout the entire process while being transformed.

Lemma 3.2 is proven in Appendix A. It implies that if the data distribution has finite second moments in the Sobolev space $H^k(\mathbb{T}^d)$, then Gaussian noise in the latent space induces a random function with the same Sobolev regularity. Consequently, the forward process of PISD obtained by gradually adding Gaussian noise, remains in $H^k(\mathbb{T}^d)$. This ensures that the PDE operators used in the PISD guidance are well defined throughout the generative process.

In contrast, the forward process of grid-based diffusion models in physical space converges, in the limit of infinite resolution, to spatial white noise, which is not differentiable. As a result, the methods proposed in Huang et al. (2024); Shu et al. (2023) apply PDE guidance only during the final 10% of the reverse process, whereas PISD enforces PDE constraints throughout. Figure 2 illustrates this difference, and our numerical results show that enforcing PDE constraints throughout enables PISD to achieve much lower PDE residuals than DiffusionPDE (cf. Section 4.1).

*Remark* 3.3 (Choice of truncation set). The truncation set $K$ determines the dimension of the latent space and hence strongly influences inference time. The canonical choice is a cube $K = \{n \in \mathbb{Z}^d : \|n\|_\infty \le c\}$ (cf. Figure 4) for some constant $c \in (0, \infty)$. To reduce dimensionality, we also experiment with approximate hyperbolic truncation $K = \{n \in \mathbb{Z}^d : |n_1 \cdots n_d| \le c\}$ (cf. Figure 3).

## 4. Numerical Results

In this section we showcase the performance of PISD on the Poisson, Helmholtz, and Navier–Stokes equations and compare it to the state-of-the-art methods DiffusionPDE (Huang et al., 2024) and FunDPS (Yao et al., 2025).

**Dataset and training.** All datasets consist of solutions and coefficients generated from the target PDEs at a resolution of $128 \times 128$ (cf. Appendix B). As denoiser architecture we use a Vision Transformer (Kolesnikov et al., 2021) that we train based on (2). We have experimented with other

architectures which all yield comparable performance.

**Comparison with other methods.** We compare PISD to the state-of-the-art diffusion-based methods Diffusion-PDE (Huang et al., 2024) and FunDPS (Yao et al., 2025). Our method specifically targets the sparse-observation regime, where classical solvers are not directly applicable and neural-operator-based methods (e.g., PINO, FNO, DeepONet) perform very poorly, as demonstrated by Huang et al. (2024); Yao et al. (2025); we therefore do not consider them direct competitors. While Tables 1 and 2 do include full-observation results, we refer the reader to Huang et al. (2024); Yao et al. (2025) for comparisons with neural operators in that regime.

**Adam guidance.** During inference, we enforce PDE and observation constraints using DPS, replacing the standard GD updates with Adam (cf. Section 3.3). We employ a frequency-aware variant of Adam, where low-frequency modes receive larger effective updates, capturing the dominant physical structure of the solution, while high-frequency modes are damped to improve numerical stability (see Appendix B.3 for a detailed description). We compare this against using GD-based guidance in DPS in Appendix C showing that our Adam-based guidance with frequency-aware weighting yields significantly better results, particularly for inverse problems.

### 4.1. Poisson and Helmholtz equations

Following Huang et al. (2024); Yao et al. (2025), we first consider partial differential equations posed on a bounded domain with homogeneous Dirichlet boundary conditions. Let $\Omega = (0, 1)^2$ and denote by $\partial\Omega$ its boundary. We consider the **Poisson equation**

$$\Delta u(x) = a(x), \quad x \in \Omega, \qquad u(x) = 0, \quad x \in \partial\Omega$$

Table 1. Forward problem, observations on $a$.

| PDE | Obs. | PISD (ours) | | DiffusionPDE | | FunDPS | |
|---|---|---|---|---|---|---|---|
| | | Rel. err | PDE res. | Rel. err | PDE res. | Rel. err | PDE res. |
| Poisson | 500 | $3.08 \pm 1.71$ % | 0.87 | $4.06 \pm 1.51$ % | 237.49 | $\mathbf{2.23 \pm 1.50}$ % | 4619.34 |
| | 1000 | $\mathbf{1.47 \pm 0.90}$ % | 2.31 | $3.35 \pm 0.99$ % | 207.28 | $1.54 \pm 1.05$ % | 3807.58 |
| | Full | $\mathbf{0.05 \pm 0.03}$ % | 3.78 | $4.04 \pm 1.50$ % | 190.70 | $0.87 \pm 0.44$ % | 3338.32 |
| Helmholtz | 500 | $3.47 \pm 1.70$ % | 0.37 | $9.55 \pm 4.16$ % | 4852.36 | $\mathbf{2.08 \pm 0.98}$ % | 3316.68 |
| | 1000 | $\mathbf{1.59 \pm 0.77}$ % | 0.52 | $7.46 \pm 2.76$ % | 4690.36 | $\mathbf{1.53 \pm 0.88}$ % | 3307.42 |
| | Full | $\mathbf{0.04 \pm 0.01}$ % | 3.97 | $8.25 \pm 3.65$ % | 5600.25 | $1.14 \pm 0.81$ % | 2714.46 |

Table 2. Inverse problem, observations on $u$.

| PDE | Obs. | PISD (ours) | | DiffusionPDE | | FunDPS | |
|---|---|---|---|---|---|---|---|
| | | Rel. err | PDE res. | Rel. err | PDE res. | Rel. err | PDE res. |
| Poisson | 500 | $\mathbf{13.81 \pm 3.11}$ % | 0.45 | $22.17 \pm 6.61$ % | 178.46 | $21.09 \pm 7.10$ % | 587.99 |
| | 1000 | $\mathbf{12.09 \pm 2.68}$ % | 0.44 | $18.14 \pm 6.04$ % | 203.62 | $20.47 \pm 6.79$ % | 460.62 |
| | Full | $\mathbf{7.95 \pm 1.71}$ % | 1.33 | $14.03 \pm 4.31$ % | 190.10 | $19.84 \pm 0.65$ % | 429.90 |
| Helmholtz | 500 | $\mathbf{12.76 \pm 2.42}$ % | 0.21 | $19.33 \pm 5.82$ % | 8916.58 | $16.26 \pm 4.46$ % | 1933.61 |
| | 1000 | $\mathbf{11.19 \pm 1.97}$ % | 0.20 | $17.03 \pm 5.08$ % | 13303.46 | $14.93 \pm 3.90$ % | 2036.62 |
| | Full | $\mathbf{9.03 \pm 2.11}$ % | 0.84 | $15.23 \pm 4.73$ % | 19010.48 | $13.97 \pm 3.60$ % | 664.21 |

Table 3. Observations on both $a$ and $u$.

| PDE | Obs. | | PISD (ours) | | DiffusionPDE | | FunDPS | |
|---|---|---|---|---|---|---|---|---|
| | | | Rel. err | PDE res. | Rel. err | PDE res. | Rel. err | PDE res. |
| Poisson | 100 | $a$ | $\mathbf{18.51 \pm 5.06}$ % | 2.11 | $18.63 \pm 6.28$ % | 230.77 | $25.49 \pm 12.57$ % | 5221.14 |
| | | $u$ | $\mathbf{1.30 \pm 0.63}$ % | | $1.39 \pm 0.74$ % | | $2.65 \pm 1.89$ % | |
| | 200 | $a$ | $\mathbf{13.38 \pm 3.67}$ % | 2.30 | $13.40 \pm 4.38$ % | 253.21 | $16.47 \pm 6.48$ % | 4573.76 |
| | | $u$ | $\mathbf{0.46 \pm 0.24}$ % | | $0.60 \pm 0.26$ % | | $1.31 \pm 0.77$ % | |
| Helmholtz | 100 | $a$ | $20.17 \pm 4.59$ % | 0.58 | $\mathbf{18.43 \pm 5.16}$ % | 12700.29 | $22.85 \pm 7.20$ % | 4823.12 |
| | | $u$ | $\mathbf{1.33 \pm 0.49}$ % | | $1.55 \pm 0.61$ % | | $2.47 \pm 1.33$ % | |
| | 200 | $a$ | $16.24 \pm 3.36$ % | 0.57 | $\mathbf{14.07 \pm 3.65}$ % | 11255.83 | $16.02 \pm 4.36$ % | 4652.91 |
| | | $u$ | $\mathbf{0.52 \pm 0.21}$ % | | $1.10 \pm 0.39$ % | | $1.45 \pm 0.58$ % | |

and the **Helmholtz equation**

$$\Delta u(x) + u(x) = a(x), \quad x \in \Omega,$$
$$u(x) = 0, \quad x \in \partial\Omega.$$

As in Example 3.1, we want to generate functions from the space $\mathcal{F} = \left(H^2(\Omega) \cap H_0^1(\Omega)\right) \times L^2(\Omega)$. As PDE residual for the Poisson equation we use the residual

$$\forall (u, a) \in \mathcal{F}: \quad \mathcal{L}_{\text{PDE}}(u, a) = \|\Delta u - a\|_{L^2(\Omega)}^2$$

and for the Helmholtz equation we use the residual

$$\forall (u, a) \in \mathcal{F}: \quad \mathcal{L}_{\text{PDE}}(u, a) = \|\Delta u + u - a\|_{L^2(\Omega)}^2.$$

To automatically enforce the Dirichlet boundary conditions, we base our encodings of $u \in H_0^2(\Omega)$ for the PISD method on the sine transform given for all $f \colon \Omega \to \mathbb{R}$, $k = (k_1, k_2) \in \mathbb{N}^2$ by

$$\hat{f}(k) = \int_\Omega f(x) \sin(\pi k_1 x_1) \sin(\pi k_2 x_2) \mathrm{d}(x_1, x_2)$$

with the corresponding inverse transform $\mathcal{I}$ based on the sine series given for all $\alpha \in \mathbb{R}^{\mathbb{N}^2}$ by

$$\sum_{k \in \mathbb{N}^2} \alpha(k) \sin(\pi k_1 x_1) \sin(\pi k_2 x_2).$$

The encoding for $a \in L^2(\Omega)$ is also based on a sine transform. Since $a$ does not satisfy zero boundary conditions, we first extend it smoothly to a larger domain on which the extended function vanishes at the boundary, then apply the sine transform. The inverse transform is obtained by evaluating the sine series and restricting to $\Omega$. To compute the PDE residual at inference time, we use the formula

$$\forall k \in \mathbb{N}^2: \quad \widehat{\Delta u}(k) = -\pi^2 \|k\|_2^2 \hat{u}(k)$$

which can be conveniently computed in terms of the latent coefficients corresponding to $u$.

**Results.** The trained models are applied to three problem classes. Table 1 reports results for *forward problems*,

Table 2 reports results for *inverse problems*, and Table 3 addresses *joint reconstruction*. The results for the forward, inverse, and joint reconstruction problem are averaged over 100 independent runs. In all cases, we draw samples from the test set and mask out parts of them to simulate sparse observations. The reported relative errors compare each reconstruction with the corresponding original, unmasked sample. PDE residuals were computed using the same finite-difference scheme for every method, including PISD (which uses spectral residuals internally during generation).

In the forward problem, PISD achieves performance comparable to DiffusionPDE and FunDPS across all observation levels, and becomes more accurate as the number of observations increases. In the inverse problem, PISD consistently outperforms DiffusionPDE and FunDPS across all observation regimes. In the joint reconstruction setting, PISD achieves the lowest error on the solution $u$ across all configurations and is the most accurate on the coefficient $a$ for the Poisson equation; on the Helmholtz equation, DiffusionPDE attains a slightly lower error on $a$.

Across all tasks, PISD also yields significantly lower PDE residuals than the baselines (cf. Figure 5 for an illustration of the PDE residuals). This suggests that the remaining error in PISD is dominated by the inherent uncertainty of the ill-posed problem under sparse observations, rather than by its inability to satisfy the PDE. This interpretation is supported by the observation that PISD's accuracy advantage over the baselines is most pronounced when more observations are available and the problem becomes less ill-posed.

Beyond matching or improving accuracy, PISD has a significantly faster inference time compared to DiffusionPDE and FunDPS due to the reduced dimensionality of the spectral latent space. From a spatial resolution of $128 \times 128$, we retain only $44 \times 44$ modes, approximately reducing inference time on a GeForce RTX 2080 Ti from 802 seconds (DiffusionPDE) and 152 seconds (FunDPS) to 52 seconds (see Table 7).

### 4.2. Navier–Stokes equations

We consider the incompressible Navier–Stokes equations in vorticity form on the periodic domain $\Omega = \mathbb{T}^2 = \mathbb{R}^2/\mathbb{Z}^2$ given by

$$\partial_t w(x,\tau) + v(x,\tau) \cdot \nabla w(x,\tau) = \nu \, \Delta w(x,\tau) + q(x),$$
$$\nabla \cdot v(x,\tau) = 0, \quad x \in \Omega, \ \tau \in (0,T].$$

Here $v = (v_1, v_2) \colon \Omega \times [0,T] \to \mathbb{R}^2$ denotes the velocity field, $w = \nabla \times v \colon \Omega \times [0,T] \to \mathbb{R}$ the vorticity, $T = 1$ the time horizon, $\nu = 0.001$ the kinematic viscosity, and $q(x_1, x_2) = 0.1(\sin(2\pi(x_1 + x_2)) + \cos(2\pi(x_1 + x_2)))$ a fixed source term.

To recover the velocity from the vorticity, we use the Biot–Savart law, which relates the two fields via their Fourier coefficients: for all $\tau \in (0,T]$, $k \in \mathbb{Z}^2 \setminus \{0\}$ we have

$$\widehat{v_1(\cdot,\tau)}(k) = i \, \frac{k_2}{\|k\|^2} \, \widehat{w(\cdot,\tau)}(k),$$
$$\widehat{v_2(\cdot,\tau)}(k) = -i \, \frac{k_1}{\|k\|^2} \, \widehat{w(\cdot,\tau)}(k). \tag{10}$$

We denote by $\mathcal{V} \colon H^2(\mathbb{T}^2) \to H^3(\mathbb{T}^2)$ the corresponding operator mapping vorticity to velocity.

We discretize the time domain into $N = 10$ steps $0 = t_1 < t_2 < \cdots < t_N = T$ and aim to generate the vorticity field $w$ on those time steps. As generation space for the PISD method we choose $\mathcal{F} = (H^2(\mathbb{T}^2))^N$, representing the vorticity at each time step. The encoding $\mathcal{E}$ and its inverse $\mathcal{I}$ are based on the complex Fourier transform applied to each time step separately as described in Section 3.4 (see in particular (6) to (8)).

Since the velocity is recovered from the vorticity via $\mathcal{V}$, we note that the divergence-free constraint $\nabla \cdot v = 0$ in the Navier–Stokes equations is automatically satisfied and does not need to be enforced in the PDE residual. Using finite differences in time, we define the PDE residual for all $w = (w_1, \ldots, w_N) \in \mathcal{F}$ by

$$\mathcal{L}_{\text{PDE}}(w)$$
$$= \sum_{i=2}^{N-1} \left\| \frac{w_{i+1} - w_{i-1}}{t_{i+1} - t_{i-1}} - \mathcal{V}(w_i) \cdot \nabla w_i - \nu \Delta w_i - q \right\|_{L^2(\mathbb{T}^2)}^2$$
$$= \sum_{i=2}^{N-1} \sum_{k \in \mathbb{Z}^2 \setminus \{0\}} \left| \frac{\widehat{w_{i+1}}(k) - \widehat{w_{i-1}}(k)}{t_{i+1} - t_{i-1}} \right.$$
$$\left. - \widehat{\mathcal{V}(w_i) \cdot \nabla w_i}(k) - \nu \|k\|_2^2 \widehat{w_i}(k) - \widehat{q}(k) \right|^2.$$

During inference, we evaluate the PDE residual using the latter equation which, except for the nonlinear advection terms $\widehat{\mathcal{V}(w_i) \cdot \nabla w_i}(k)$, is conveniently expressed in terms of the latent coefficients. The terms $\widehat{\mathcal{V}(w_i) \cdot \nabla w_i}(k)$ are computed via a pseudo-spectral method: the spatial derivative of $\nabla w_i$ is computed explicitly in Fourier space and the Fourier coefficients of $\mathcal{V}(w_i)$ are computed using the Biot–Savart law in (10), both $\mathcal{V}(w_i)$ and $\nabla w_i$ are transformed to physical space for pointwise multiplication, and the result is then transformed back to Fourier space.

*Table 4.* Navier–Stokes with sparse-in-time observations.

| Obs. | Time | Data | PISD (ours) | | DiffusionPDE | FunDPS |
|---|---|---|---|---|---|---|
| | | | Rel. err | PDE res. | Rel. err | Rel. err |
| 500 | $t_1$ | ✓ | **5.19 ± 0.93 %** | – | 6.55 ± 1.16 % | 7.49 ± 1.37 % |
| | $t_2$ | – | 4.27 ± 0.79 % | 0.14 | – | – |
| | $t_3$ | – | 3.45 ± 0.52 % | 0.15 | – | – |
| | $t_4$ | – | 3.47 ± 0.54 % | 0.14 | – | – |
| | $t_5$ | – | 3.08 ± 0.46 % | 0.10 | – | – |
| | $t_6$ | – | 3.02 ± 0.47 % | 0.11 | – | – |
| | $t_7$ | – | 2.48 ± 0.41 % | 0.14 | – | – |
| | $t_8$ | – | 2.05 ± 0.34 % | 0.15 | – | – |
| | $t_9$ | – | 1.65 ± 0.38 % | 0.15 | – | – |
| | $t_{10}$ | ✓ | **0.21 ± 0.06 %** | – | 0.50 ± 0.09 % | 1.16 ± 0.22 % |
| 500 | $t_1$ | – | 4.24 ± 0.62 % | – | – | – |
| | $t_2$ | – | 2.42 ± 0.36 % | 0.04 | – | – |
| | $t_3$ | – | 2.00 ± 0.32 % | 0.08 | – | – |
| | $t_4$ | ✓ | 0.77 ± 0.18 % | 0.04 | – | – |
| | $t_5$ | – | 1.18 ± 0.20 % | 0.08 | – | – |
| | $t_6$ | – | 1.06 ± 0.21 % | 0.08 | – | – |
| | $t_7$ | ✓ | 0.32 ± 0.09 % | 0.05 | – | – |
| | $t_8$ | – | 1.72 ± 0.31 % | 0.07 | – | – |
| | $t_9$ | – | 1.49 ± 0.33 % | 0.04 | – | – |
| | $t_{10}$ | – | 2.61 ± 0.49 % | – | – | – |
| 200 | $t_1$ | ✓ | **8.80 ± 1.38 %** | – | 10.30 ± 1.88 % | 11.32 ± 2.06 % |
| | $t_2$ | – | 7.08 ± 1.11 % | 0.13 | – | – |
| | $t_3$ | – | 5.96 ± 0.90 % | 0.19 | – | – |
| | $t_4$ | – | 5.14 ± 0.78 % | 0.15 | – | – |
| | $t_5$ | – | 4.46 ± 0.69 % | 0.09 | – | – |
| | $t_6$ | – | 3.93 ± 0.61 % | 0.10 | – | – |
| | $t_7$ | – | 3.27 ± 0.55 % | 0.16 | – | – |
| | $t_8$ | – | 2.63 ± 0.47 % | 0.19 | – | – |
| | $t_9$ | – | 2.14 ± 0.43 % | 0.13 | – | – |
| | $t_{10}$ | ✓ | **1.57 ± 0.43 %** | – | 1.84 ± 0.45 % | 2.74 ± 0.66 % |

**Results.** Table 4 reports results averaged over 100 independent runs for generating the full spatio-temporal evolution of the vorticity field at time steps $t_1, \ldots, t_{10}$, conditioned on sparse observations at various times. As before, we draw ground-truth trajectories from the test set, mask out parts of them to simulate sparse observations, and report relative errors against the corresponding unmasked trajectories.

Unlike DiffusionPDE and FunDPS, which generate only initial or terminal states, PISD generates the full trajectory and supports conditioning on initial, final, or intermediate observations. This is enabled by the fast inference afforded by the spectral representation: from a spatial resolution of $128 \times 128$, we retain only $32 \times 32$ Fourier modes per time step.

Run times are reported in Table 8. In the first setting of Table 4, PISD achieves better accuracy on the initial and terminal states than both baselines—at less than twice the runtime of FunDPS and half that of DiffusionPDE—while additionally generating all intermediate time steps.

## 5. Conclusion

We have introduced *physics-informed spectral diffusion* (PISD), a generative framework for parametric PDEs that operates in a latent space of scaled spectral coefficients and enforces physics-informed constraints during inference using Adam-based updates. By normalizing spectral coefficients according to the data distribution, PISD ensures that the diffusion process remains within a class of functions

with controlled Sobolev regularity, allowing PDE guidance throughout the sampling process and yielding significantly lower PDE residuals than existing methods.

Across forward and inverse problems for Poisson, Helmholtz, and Navier–Stokes equations, PISD matches or outperforms existing diffusion-based PDE solvers in reconstruction accuracy while significantly reducing inference time, achieving roughly a 3× speedup compared to FunDPS and a 15× speedup compared to DiffusionPDE.

Overall, PISD provides a physically grounded and computationally efficient approach for generative PDE modeling.

## 6. Limitations and Future Work

PISD relies on a spectral representation that must be specified for each PDE, and is naturally suited to regular domains with periodic, homogeneous Dirichlet, or Neumann boundary conditions. Extending the approach to irregular geometries or more general boundary conditions would require techniques such as domain decomposition or alternative function bases. A promising direction for future work is to replace hand-crafted spectral encodings with learned encoder-decoder pairs, for instance using neural operators as in FunDiff (Wang et al., 2025), which could enable automatic adaptation to diverse problem settings. Additionally, the current framework requires a dataset of solution fields for training. Extending to low-data regimes is an important direction for future work.

## Impact Statement

This paper presents work whose goal is to advance the field of Machine Learning. There are many potential societal consequences of our work, none of which we feel must be specifically highlighted here.

## Acknowledgments

This work has been partially funded by the National Science Foundation of China (NSFC) under grant number W2531010. Calculations (or parts of them) for this publication were performed on the HPC cluster PALMA II of the University of Münster, subsidised by the DFG (INST 211/667-1). Financial support from Swiss National Science Foundation Grant 10003723 is gratefully acknowledged. Moreover, we gratefully acknowledge the Cluster of Excellence EXC 2044/2–390685587, Mathematics Münster: Dynamics-Geometry-Structure funded by the Deutsche Forschungsgemeinschaft (DFG, German Research Foundation).

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

# A. Proof of Lemma 3.2

*Proof.* The proof of Lemma 3.2 relies on the following elementary connection between Sobolev spaces and Fourier coefficients (cf., e.g., Lemma 5.4 in Einsiedler & Ward (2017)). We have that

$$H^k(\mathbb{T}^d) = \left\{ f \in L^2(\mathbb{T}^d) \colon \ \sum_{n \in \mathbb{Z}^d} |\hat{f}(n)|^2 \|n\|_2^{2k} < \infty \right\}$$

and for all $f \in H^k(\mathbb{T}^d)$ the Sobolev norm is equivalent to

$$\|f\|_{H^k(\mathbb{T}^d)}^2 \simeq \sum_{n \in \mathbb{Z}^d} |\hat{f}(n)|^2 \|n\|_2^{2k}. \tag{11}$$

Note that (9) and the fact that $(\phi_n)_{n \in \mathbb{Z}^d}$ is an orthonormal basis of $L^2(\mathbb{T}^d)$ implies that for all $n \in \mathbb{Z}^d$ we have that

$$\hat{N}(n) = s_n Z_n.$$

The assumption that $\mathbb{E}\big[\|X_{\text{data}}\|_{H^k(\mathbb{T}^d)}^2\big] < \infty$, (7), and (11) therefore imply that

$$
\begin{aligned}
\mathbb{E}\Big[\|N\|_{H^k(\mathbb{T}^d)}^2\Big] &\simeq \mathbb{E}\left[ \sum_{n \in \mathbb{Z}^d} |\hat{N}(n)|^2 \|n\|_2^{2k} \right] \\
&= \mathbb{E}\left[ \sum_{n \in \mathbb{Z}^d} |s_n Z_n|^2 \|n\|_2^{2k} \right] \\
&= \sum_{n \in \mathbb{Z}^d} (s_n)^2 \|n\|_2^{2k}\, \mathbb{E}\big[|Z_n|^2\big] \\
&= \sum_{n \in \mathbb{Z}^d} \text{Var}\big(\widehat{X_{\text{data}}}(n)\big) \|n\|_2^{2k} \\
&\le \sum_{n \in \mathbb{Z}^d} \mathbb{E}\Big[|\widehat{X_{\text{data}}}(n)|^2\Big] \|n\|_2^{2k} \\
&= \mathbb{E}\left[ \sum_{n \in \mathbb{Z}^d} |\widehat{X_{\text{data}}}(n)|^2 \|n\|_2^{2k} \right] \\
&\simeq \mathbb{E}\Big[\|X_{\text{data}}\|_{H^k(\mathbb{T}^d)}^2\Big] < \infty.
\end{aligned}
$$

$\square$

# B. Experiment description

### B.1. Dataset and spectral transform preprocessing

All experiments use the datasets from DiffusionPDE (Huang et al., 2024), generated numerically at a spatial resolution of $128 \times 128$. For the Poisson and Helmholtz equations, each sample consists of a (PDE coefficient, solution) pair obtained via a second-order finite difference scheme. For the Navier–Stokes equations, samples are vorticity trajectories generated following the FNO framework (Li et al., 2021) using its publicly released code; each trajectory consists of 10 consecutive time steps, corresponding to one second of physical simulation time, and constitutes a complete spatio-temporal snapshot of the vorticity field over that interval. We use the original training and test splits provided in DiffusionPDE to ensure a fair and direct comparison with existing diffusion-based PDE solvers.

Prior to training, all fields are transformed into the spectral domain. For the Poisson and Helmholtz equations, we employ a discrete sine transform, while for the Navier–Stokes equations we use a standard Fourier transform consistent with the periodic boundary conditions. Note that for the Poisson and Helmholtz equations, the discrete sine transform requires the

input to vanish at the boundary. The PDE coefficients do not satisfy this requirement, so applying the sine transform directly would be ill-posed and produce Gibbs-type spectral artifacts. To address this, we pad the coefficients $a$ with four external layers that decrease smoothly to zero at the boundary, making the sine transform well-defined.

To reduce the effective dimensionality and focus learning on the dominant modes, we truncate the spectral coefficients. All subsequent training and inference operations are performed in this truncated spectral space. For Poisson and Helmholtz, we retain $44 \times 44$ modes using a hyperbolic truncation strategy (see Figure 3), which prioritizes low-frequency modes while gradually reducing high-frequency components. To produce a fixed-size square input for the network, the axis-aligned strips are folded into the remaining entries.

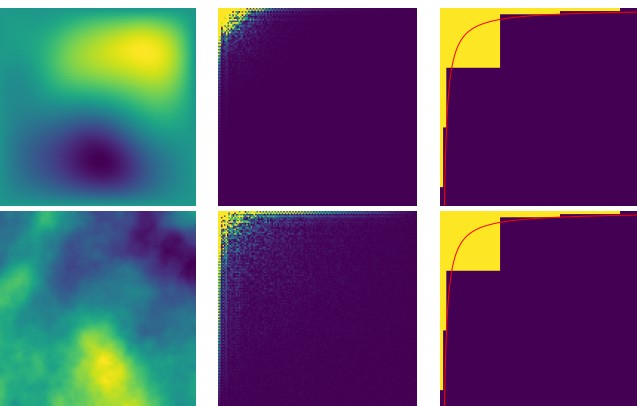

*Figure 3.* Truncation of sine coefficients used for the Poisson and Helmholtz equations illustrated on a Poisson example. Left: heatmap of the solution $u$ and corresponding coefficient $a$. Center: a heatmap of the sine coefficients shown on a clipped color scale to make the smaller high-frequency components visible. Right: the hyperbolic truncation mask with the boundary highlighted in red.

For Navier–Stokes equations, we retain the inner $32 \times 32$ square of Fourier modes (selected via fftshift prior to truncation), which, after inverting the shift, results in Figure 4.

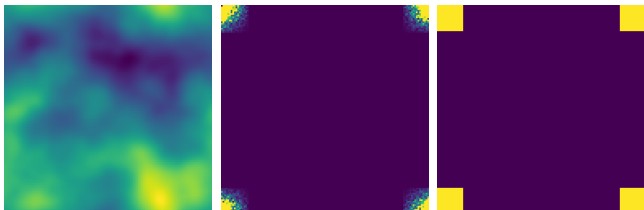

*Figure 4.* Truncation of Fourier coefficients for the Navier–Stokes problem. Left: the vorticity field $w$ at the initial time $t_1$. Center: the magnitude of its Fourier coefficients shown without an fftshift so that low frequencies appear in the four corners. Right: the truncation mask that retains the inner $32 \times 32$ block of low-frequency modes (yellow corners after inverse shift correspond to a centered low-frequency block).

We have chosen the spectral truncation so that the reconstructed fields closely match the original solutions. We ablate the truncation strategy in Appendix C.3, confirming that our reduction of the full $128 \times 128$ resolution to a $44 \times 44$ coefficient grid for Poisson and Helmholtz and $32 \times 32$ for Navier–Stokes provides a good accuracy/cost trade-off.

### B.2. Training details

All training runs were performed on 4 NVIDIA GeForce RTX 2080 GPUs. Training time for the Poisson and Helmholtz datasets was approximately 3 hours, while training for the Navier–Stokes dataset required about 10 hours. Although the per-frame spectral truncation for Navier–Stokes is smaller ($32 \times 32$ versus $44 \times 44$), each sample comprises a full trajectory of 10 time steps, substantially increasing the effective problem dimensionality and thus the training cost compared to the Poisson and Helmholtz equations.

The network architecture for the denoiser is based on a Vision Transformer. We also experimented with a U-Net architecture,

which provided comparable results in terms of accuracy. In all cases, our model has 2M parameters compared to the 54M in the DiffusionPDE and FunDPS networks.

The training objective is a standard denoising diffusion loss applied to the truncated coefficients.

### B.3. Inference details

For the Poisson, Helmholtz, and Navier–Stokes equations, we perform 100 independent simulations in each experimental setting. The principal metric we report is the relative error, while we also evaluate the PDE residual to quantify how well the generated solutions satisfy the underlying differential equations.

Our method consistently produces lower PDE residuals compared to baseline approaches, thanks to the accurate computation of derivatives in Fourier space. For example, in Figure 5 we compare the Laplacian of the solution $u$ computed with our Fourier-based derivatives versus finite-difference approximations obtained in other methods. The figure illustrates that our approach captures the differential structure more accurately, which directly contributes to improved PDE consistency.

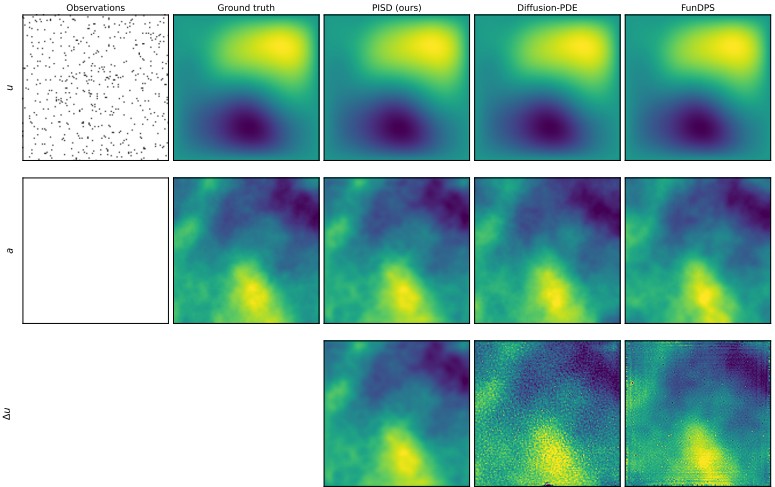

*Figure 5.* Inverse Poisson problem with 500 sparse observations of $u$ and no observations of $a$. Top row: the observation mask, the ground truth, and the reconstructed solutions $u$. Middle row: the empty observation mask, the ground truth, and the reconstructed coefficients $a$. Bottom row: the Laplacian $\Delta u$ of each reconstruction, for the Poisson equation, this should match the coefficient $a$ in the middle row. PISD computes $\Delta u$ spectrally and recovers a clean Laplacian, whereas the finite-difference Laplacians of DiffusionPDE and FunDPS exhibit substantial noise.

**Guidance coefficients.**  The guidance coefficients used during inference vary slightly depending on the task and number of observations. We report the values for the Poisson and Helmholtz equations in Table 5 and for the Navier–Stokes equations in Table 6. We refer to Appendix C.2 for an ablation of the guidance coefficients.

*Table 5.* Guidance coefficients used for Poisson and Helmholtz problems.

| Case | Obs. | $\lambda_u$ | $\lambda_a$ | $\lambda_{\text{PDE}}$ |
|---|---|---|---|---|
| Forward problem | 500 | 0 | 0.05 | 0.0005 |
| | 1000 | 0 | 0.05 | 0.0005 |
| | Full | 0 | 0.05 | 0.0001 |
| Inverse problem | 500 | 20 | 0 | 0.00005 |
| | 1000 | 20 | 0 | 0.00005 |
| | Full | 40 | 0 | 0.000005 |
| Joint reconstruction | 100 | 40 | 0.05 | 0.0002 |
| | 200 | 40 | 0.05 | 0.0002 |

*Table 6.* Guidance coefficients used for the Navier–Stokes equations.

| Obs. | $\lambda_{obs}$ | $\lambda_{\text{PDE}}$ |
|---|---|---|
| 200 | 0.0001 | 0.5 |
| 500 | 0.00001 | 0.5 |

**Inference time.** Table 7 reports the time required to generate a single solution on a GeForce RTX 2080 Ti for the Poisson and Helmholtz equations, comparing PISD against DiffusionPDE and FunDPS. For the Navier–Stokes equations, a direct comparison is less meaningful since PISD generates the full temporal trajectory, whereas the baseline methods produce only the initial and final states. We nonetheless report absolute runtimes in Table 8 for completeness.

*Table 7.* Time (in seconds) to generate one solution on a Geforce RTX 2080 Ti.

| PDE | PISD (ours) | DiffusionPDE | FunDPS |
|---|---|---|---|
| Poisson | 52 | 802 | 153 |
| Helmholtz | 52 | 802 | 171 |

*Table 8.* Runtime comparison for the Navier–Stokes equations.

| Method | Runtime (s) |
|---|---|
| PISD (full trajectory) | 420 |
| DiffPDE (initial and terminal state) | 809 |
| FunDPS (initial and terminal state) | 246 |

**Frequency-aware Adam guidance.** During inference, PDE and observation constraints are enforced via gradient-based guidance based on DPS (see Section 3.3). Unlike standard DPS, which typically uses plain GD, we employ a frequency-aware variant of the Adam optimizer. The first- and second-moment estimates are maintained across diffusion steps, and updates are modulated by a frequency-dependent learning rate to prioritize physically meaningful low-frequency modes.

Specifically, let $\hat{X}_k$ denote the Fourier coefficient at mode $k$ and $g_k$ the gradient. The update at step $t$ is computed as:

$$m_k^{(t)} = \beta_1 \, m_k^{(t-1)} + (1 - \beta_1)g_k,$$
$$v_k^{(t)} = \beta_2 \, v_k^{(t-1)} + (1 - \beta_2)g_k^2,$$
$$\hat{X}_k^{(t+1)} = \hat{X}_k^{(t)} - \eta_k \frac{m_k^{(t)}}{\sqrt{v_k^{(t)} + \epsilon}},$$

where the effective learning rate $\eta_k$ depends on the frequency mode:

$$\eta_k = \begin{cases} \text{lr}_{\text{low}}, & \text{for low frequencies} \\ \text{lr}_{\text{high}}, & \text{for high frequencies,} \end{cases}$$

with a smooth interpolation between the two values over a transition band at intermediate frequencies. This formulation allows the low-frequency modes to be updated aggressively, capturing the main structure of the solution, while high-frequency modes are updated conservatively to reduce noise amplification. This design is important for stabilizing the PDE- and observation-constrained inference process in PISD and achieving low PDE residuals in all tested scenarios. We list the concrete choices of parameters in our experiments in Table 9.

*Table 9.* Frequency-aware Adam hyperparameters used across experimental settings.

| Problem | PDE | $\beta_1$ | $\beta_2$ | $lr_{low}$ | $lr_{high}$ |
|---|---|---|---|---|---|
| Forward & Inverse - partial observation | Poisson & Helmholtz | 0.985 | 0.98 | 0.2 | 0.01 |
| Forward & Inverse - full observation | Poisson & Helmholtz | 0.97 | 0.98 | 0.2 | 0.01 |
| Joint reconstruction | Poisson & Helmholtz | 0.985 | 0.98 | 0.1 | 0.01 |
| All scenarios | Navier–Stokes | 0.97 | 0.98 | 0.1 | 0.01 |

# C. Ablation studies

## C.1. Ablations over optimization techniques

We assess the impact of the optimization technique used in the inference phase of PISD through three ablations on the Poisson equation. First, we show on FunDPS that replacing plain GD with adaptive optimizers improves DPS-based guidance. Second, we jointly ablate the choice of optimizer (GD, Momentum, Adam) and the frequency-aware learning-rate schedule within PISD. Third, we study the robustness of the best configuration (frequency-aware Adam) against the plain GD baseline across varying numbers of observations and both forward and inverse problems. Together, the ablations reported in Tables 10–12 support the frequency-aware Adam optimizer as the preferred optimization technique for DPS-based guidance in PISD.

**FunDPS baseline.** We verify that the benefit of adaptive optimization in the enforcement of constraints is not specific to PISD. Table 10 reports the relative error on the inverse Poisson problem for FunDPS with different gradient-based guidance. Notably, Adam achieves the lowest relative error, outperforming both plain GD and Momentum, which confirms that adaptive optimization is a meaningful design choice for DPS-based diffusion guidance and that this advantage is not specific to PISD. We also refer to Belardi et al. (2026) who have concurrently and independently proposed the same Adam-based replacement in DPS guidance and have validated it extensively on image generation tasks.

*Table 10.* FunDPS with different gradient-based guidance on the inverse Poisson problem (500 observations).

| Method | Rel. err. |
|---|---|
| FunDPS + GD | $21.09 \pm 7.10\,\%$ |
| FunDPS + Momentum | $18.93 \pm 5.67\,\%$ |
| FunDPS + Adam | $17.44 \pm 4.79\,\%$ |

**Optimizer and frequency-aware learning rate within PISD.** Table 11 reports a joint ablation of the optimizer and the frequency-aware learning-rate schedule within PISD. Adam is consistently the best optimizer for DPS-based guidance, and the frequency-aware schedule yields an additional gain on the inverse problem while remaining competitive in the forward case.

*Table 11.* Optimizer ablation in PISD applied to the Poisson problem (500 observations).

| Method | Inverse problem | | Forward problem | |
|---|---|---|---|---|
| | Freq.-aware | Standard | Freq.-aware | Standard |
| PISD + GD | $49.25 \pm 15.42\%$ | $61.11 \pm 17.90\%$ | $3.60 \pm 3.04\%$ | $3.62 \pm 1.97\%$ |
| PISD + Momentum | $50.10 \pm 14.45\%$ | $42.74 \pm 12.97\%$ | $3.08 \pm 2.25\%$ | $3.10 \pm 2.31\%$ |
| PISD + Adam (ours) | $\mathbf{13.81 \pm 3.11}\%$ | $17.38 \pm 3.75\%$ | $\mathbf{3.08 \pm 1.71}\%$ | $3.82 \pm 2.30\%$ |

**Robustness across observation regimes.** Finally, we compare the two extremes, plain GD and our frequency-aware Adam, across varying numbers of observations and both forward and inverse problems (Table 12). Frequency-aware Adam consistently achieves lower relative error, lower PDE residuals, and substantially lower variance. The gap is particularly pronounced in the inverse problem, where plain GD frequently fails to produce meaningful solutions.

*Table 12.* Frequency-aware Adam vs. plain GD guidance in PISD applied to the Poisson problem.

| Case | Obs. | Freq.-aware Adam (ours) | | | Plain GD | | |
|---|---|---|---|---|---|---|---|
| | | Rel. err. | PDE res. | Obs. rel. err. | Rel. err. | PDE res. | Obs. rel. err. |
| Forward | 500 | $3.08 \pm 1.71\,\%$ | 0.87 | $0.11\,\%$ | $3.60 \pm 3.04\,\%$ | 12.35 | $10.99\,\%$ |
| | 1000 | $1.47 \pm 0.90\,\%$ | 2.31 | $0.10\,\%$ | $2.12 \pm 1.34\,\%$ | 12.66 | $12.30\,\%$ |
| | Full | $0.05 \pm 0.03\,\%$ | 3.78 | $2.38\,\%$ | $0.90 \pm 0.43\,\%$ | 13.26 | $13.55\,\%$ |
| Inverse | 500 | $13.81 \pm 3.11\,\%$ | 0.45 | $0.12\,\%$ | $49.25 \pm 15.42\,\%$ | 7.73 | $5.85\,\%$ |
| | 1000 | $12.09 \pm 2.68\,\%$ | 0.44 | $0.12\,\%$ | $50.52 \pm 15.56\,\%$ | 19.65 | $6.28\,\%$ |
| | Full | $7.95 \pm 1.71\,\%$ | 1.33 | $0.10\,\%$ | $50.41 \pm 15.59\,\%$ | 22.23 | $6.02\,\%$ |

## C.2. Ablation over guidance weights

We assess the robustness of PISD to the guidance weights $\lambda_{\mathrm{obs}}$ and $\lambda_{\mathrm{PDE}}$ on the Poisson problem with 500 observations, varying each weight independently around its default. Results for the forward and inverse settings are reported in Tables 13 and 14, respectively. The relative error is remarkably stable across several orders of magnitude in both regimes, which we attribute to the adaptive step sizes of the Adam optimizer that automatically rescale the contribution of each guidance term. PISD therefore does not require careful per-task tuning of $\lambda_a$, $\lambda_u$, and $\lambda_{\mathrm{PDE}}$.

*Table 13.* Sensitivity of PISD to $\lambda_a$ and $\lambda_{\mathrm{PDE}}$ in case of the forward Poisson problem (500 observations on $a$).

| | Varying $\lambda_a$ | | Varying $\lambda_{\mathrm{PDE}}$ |
|---|---|---|---|
| $\lambda_a$ | Rel. err. | $\lambda_{\mathrm{PDE}}$ | Rel. err. |
| $10^{-4}$ | $3.96 \pm 2.21\,\%$ | $10^{-5}$ | $3.67 \pm 2.24\,\%$ |
| $10^{-3}$ | $3.15 \pm 1.66\,\%$ | $10^{-4}$ | $3.40 \pm 1.94\,\%$ |
| $10^{-2}$ | $2.96 \pm 1.72\,\%$ | $10^{-3}$ | $3.19 \pm 1.95\,\%$ |
| $10^{-1}$ | $3.43 \pm 1.73\,\%$ | $10^{-2}$ | $2.96 \pm 1.57\,\%$ |
| $10^{0}$ | $3.48 \pm 2.06\,\%$ | $10^{-1}$ | $3.89 \pm 1.89\,\%$ |

*Table 14.* Sensitivity of PISD to $\lambda_u$ and $\lambda_{\mathrm{PDE}}$ in case of the inverse Poisson problem (500 observations on $u$).

| | Varying $\lambda_u$ | | Varying $\lambda_{\mathrm{PDE}}$ |
|---|---|---|---|
| $\lambda_u$ | Rel. err. | $\lambda_{\mathrm{PDE}}$ | Rel. err. |
| $10^{0}$ | $19.30 \pm 4.30\,\%$ | $10^{-8}$ | $18.27 \pm 5.72\,\%$ |
| $10^{1}$ | $15.75 \pm 3.31\,\%$ | $10^{-7}$ | $11.67 \pm 3.18\,\%$ |
| $10^{2}$ | $11.86 \pm 2.66\,\%$ | $10^{-6}$ | $11.77 \pm 2.72\,\%$ |
| $10^{3}$ | $11.73 \pm 3.39\,\%$ | $10^{-5}$ | $15.86 \pm 3.34\,\%$ |
| $10^{4}$ | $18.18 \pm 5.56\,\%$ | $10^{-4}$ | $19.34 \pm 4.37\,\%$ |

## C.3. Ablation over truncation dimension

Our default configuration for Poisson and Helmholtz problems retains $44 \times 44$ sine coefficients with a hyperbolic truncation strategy (see Figure 3). To assess the sensitivity of PISD to the choice of truncation strategy, we retrain the model on the Poisson problem under two additional hyperbolic truncation strategies ($32 \times 32$, $64 \times 64$) and a square truncation ($44 \times 44$), which retains the full square block of modes without any hyperbolic technique (see Figure 6). Results are reported in Table 15.

*Table 15.* Effect of the truncation strategy in PISD applied to the Poisson problem (500 observations). Default in **bold**.

| Modes | Strategy | Forward rel. err. (%) | Inverse rel. err. (%) |
|---|---|---|---|
| $32 \times 32$ | Hyperbolic | $3.17 \pm 1.81\,\%$ | $20.05 \pm 3.17\,\%$ |
| **$44 \times 44$** | **Hyperbolic** | **$3.08 \pm 1.71$** % | **$13.81 \pm 3.11$** % |
| $64 \times 64$ | Hyperbolic | $3.69 \pm 2.02\,\%$ | $12.03 \pm 3.41\,\%$ |
| $44 \times 44$ | Square | $4.99 \pm 4.17\,\%$ | $28.94 \pm 2.73\,\%$ |

We observe two trends. First, performance is stable across hyperbolic strategies: the smaller $32 \times 32$ model is competitive, while the larger $64 \times 64$ model offers only marginal gains at the cost of slower training and inference. This indicates that the dominant frequency content of the Poisson solutions is already well captured at $44 \times 44$, justifying our default choice as a good accuracy/cost trade-off. Second, at matched nominal resolution, hyperbolic truncation outperforms square truncation, confirming that keeping high-frequency modes improves reconstruction quality and derivative accuracy.

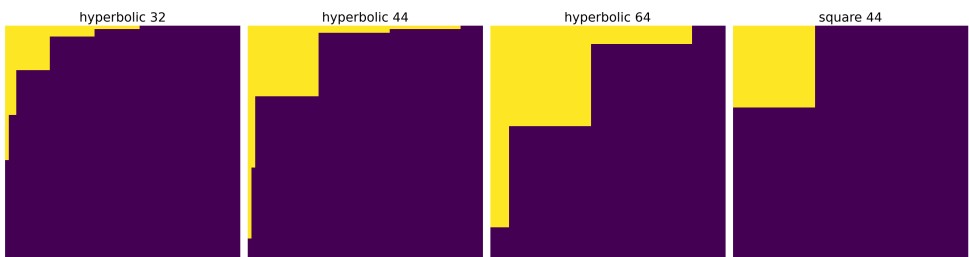

*Figure 6.* Truncation masks applied to the sine transform coefficients. Yellow regions indicate retained modes. From left to right: hyperbolic masks with resolution $32 \times 32$, $44 \times 44$, and $64 \times 64$, followed by a square $44 \times 44$ mask. The hyperbolic masks retain a low-frequency core together with axis-aligned strips of higher-frequency modes. To obtain the square representation, these axis-aligned strips are folded into the remaining entries of the square so that all retained modes fit into a compact array suitable as network input. The square mask, by contrast, retains all modes in a uniform block, discarding the high-frequency terms that the hyperbolic strategy retains.

## D. Additional figures

**Inference process figures.** In Figure 7 we provide a comparison of the inference process between our method and the DiffusionPDE paper.

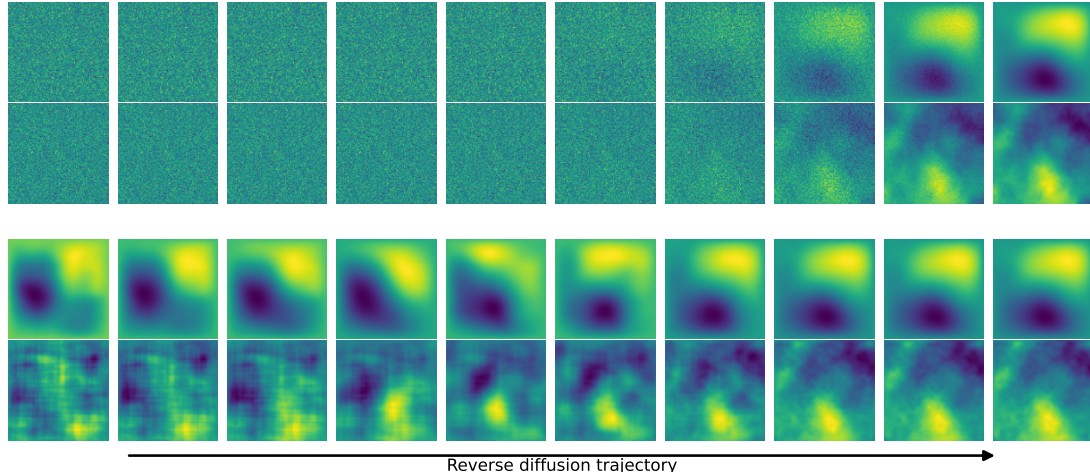

Reverse diffusion trajectory

*Figure 7.* Heatmaps of the solution $u$ and coefficient $a$ along the reverse diffusion process for the Poisson problem. Top block (DiffusionPDE): the first row shows $u$ and the second shows $a$; both fields evolve in physical space and remain noisy until the last denoising steps. Bottom block (PISD, ours): same layout, but with fields obtained by decoding the spectral latent at each step. Throughout the entire trajectory, PISD's iterates remain in a regular function class (cf. Lemma 3.2).

**Qualitative results.** We present qualitative examples of solutions generated by PISD, DiffusionPDE, and FunDPS under sparse observation regimes. Figures 8 and 9 illustrate forward and inverse problems for Poisson and Helmholtz equations with 500 observations, showing reconstructions of both the solution $u$ and the coefficient $a$, together with the corresponding pointwise error maps.

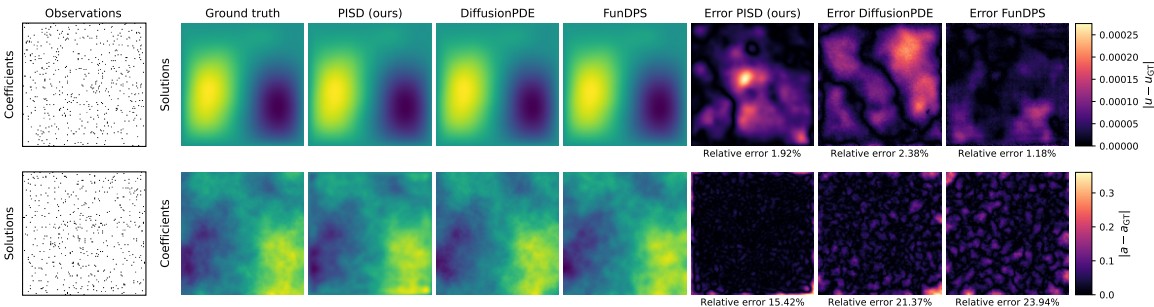

*Figure 8.* Heatmaps of reconstructions for the Poisson forward problem (top) with 500 sparse observations of the coefficient $a$, and the Poisson inverse problem (bottom) with 500 sparse observations of the solution $u$. For each task, we show the observation mask, the ground truth, the reconstructions produced by PISD, DiffusionPDE, and FunDPS, and the corresponding pointwise absolute errors. Relative errors are reported below each error map.

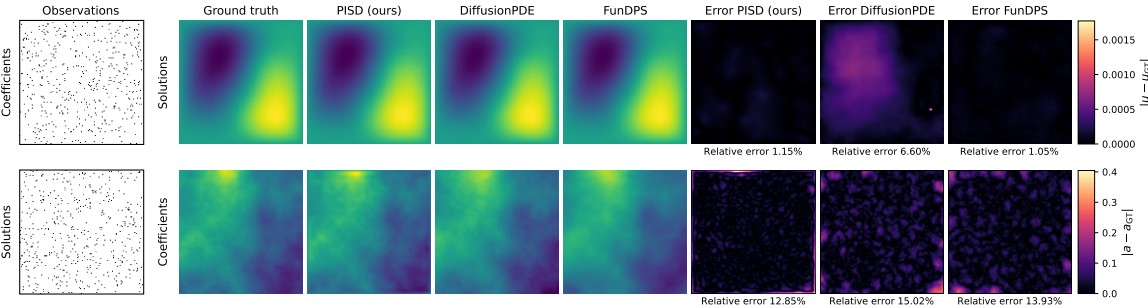

*Figure 9.* Heatmaps of reconstructions for the Helmholtz forward problem (top) with 500 sparse observations of the coefficient $a$, and the Helmholtz inverse problem (bottom) with 500 sparse observations of the solution $u$. For each task, we show the observation mask, the ground truth, the reconstructions produced by PISD, DiffusionPDE, and FunDPS, and the corresponding pointwise absolute errors. Relative errors are reported below each error map.

Figure 10 shows a Navier–Stokes example conditioned only on sparse observations at the first and last time steps. The model successfully reconstructs the full spatio-temporal evolution of the flow, producing coherent intermediate dynamics that satisfy the governing equations. These qualitative results complement the quantitative comparisons and highlight the ability of PISD to enforce PDE constraints while maintaining global consistency under sparse supervision.

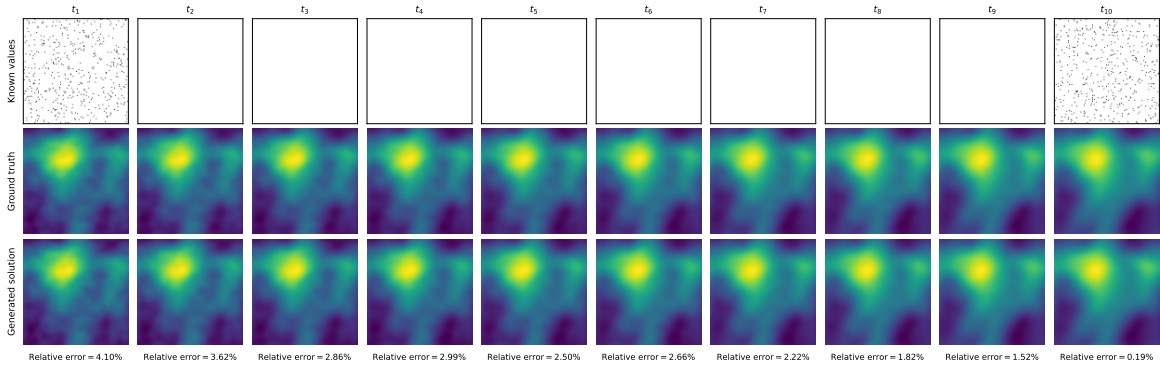

*Figure 10.* Heatmaps of the vorticity field $w$ along a Navier–Stokes trajectory reconstructed by PISD from 500 sparse observations at the initial time $t_1$ and the final time $t_{10}$ (no observations at intermediate times). Top row: observation masks (only $t_1$ and $t_{10}$ contain known values). Middle row: ground-truth vorticity. Bottom row: PISD reconstruction. Relative errors at each time step are reported below the corresponding column. PISD reconstructs the full spatio-temporal evolution.

