# OpenReview forum: "Physics-Informed Diffusion Models in Spectral Space"
_ICML.cc/2026/Conference — ICML 2026 regular_

### Official Review · Reviewer_y1fm · 2026-02-27

**Soundness:** 2
**Presentation:** 3
**Significance:** 2
**Originality:** 2
**Overall Recommendation:** 4
**Confidence:** 2

**Summary:**

The authors propose a diffusion model approach that operates in a reduced spectral space to solve sparse-observation PDEs, which they term by Physics-Informed Spectral Diffusion (PISD).

**Compliance With Llm Reviewing Policy:**

Affirmed.

**Final Justification:**

I would like to thank the authors for their response. Most of my major concerns are addressed. Therefore, I will increase my score and vote to accept the paper.

**Key Questions For Authors:**

See weaknesses.

**Limitations:**

The authors do include a limitation section. However, it is mostly future work.

**Strengths And Weaknesses:**

## Strengths:

- The paper is, in general, well-written and easy to follow.

- The spectral encoding, its justification, and the resulted accelerated convergence.


## Weaknesses/Questions:

- Why only considering the sparse-regime? Comparison with NOs is needed in the setup where NOs operate well. This is to determine the capabilities and limitations of the proposed sampler.

- How is the performance when compared to classical finite element methods? . This is also to determine the capabilities and limitations of the proposed sampler.

- Robustness to small-parameter change and OOD instances would strengthen the proposed approach.

- Why Adam is compared to SGD? Where would the stochasticity in batches come from in step 10? Could the ADAM significant improvement be coming from using momentum parameters? In this case, a comparison with momentum-based GD is needed (MGD). The second moment may not be needed. MGD is a much lighter optimizer that could result in further speed ups.

- Why are the run-time results in Table 7 only w.r.t. two problems? Is acceleration achieved on the remaining problems and across different observation regimes?

- How is the performance sensitivity to the choice of lambda_obs and lambda_PDE across different problems and observations?



## Minor:

- X and Y labels in Figure 2 and the many other Appendix figures are needed for better readability.

- It is strange to use an ordered steps between the training and testing procedures in the Algorithm.

---

> ### Author Rebuttal · Authors · 2026-03-31
>
> We thank the reviewer for the constructive feedback.
>
> > **Why only considering the sparse-regime? (Q1)**
>
> Thank you for the suggestion. Our method specifically targets the sparse-observation regime, where traditional solvers are not directly applicable and NOs perform very poorly, as shown by DiffPDE and FunDPS. We thus do not consider NOs direct competitors. While Tables 1 and 2 do include full-observation results, we refer the reader to DiffPDE and FunDPS for comparisons with NOs in that regime.
>
> > **Comparisons with classical finite element methods (Q2)**
>
> Traditional solvers can provide solutions of arbitrary precision given full PDE specifications but are not designed for the sparse-observation setting, which is our main focus. The full-observation regime is a limit case of our framework, but we do not attempt to beat traditional solvers in this case. Note that this is consistent with the evaluation protocol of DiffPDE and FunDPS, which also omit such comparisons.
>
> > **Robustness to small-parameter change and OOD instances (Q3)**
>
> Thanks for the question. The training distribution has support over the entire PDE solution space, so any valid instance is in-distribution by construction and there are no true OOD instances.
>
> If you mean small changes in the PDE parameters (e.g., a different viscosity), PISD could conceivably still work without retraining by adjusting the PDE loss at inference, but this is beyond the intended use case of our method and the baselines.
>
> > **Comparison of Adam with SGD and momentum-based GD (Q4)**
>
> Thank you for your observation regarding SGD. There is indeed no stochasticity in step 10; the comparison is between Adam and deterministic GD. We will correct this typo in Table 8.
>
> In response to your suggestion, we have conducted additional ablations comparing GD, MGD, and Adam - each with and without frequency-aware learning rates - on the Poisson problem with 500 observations (inverse problem first, forward problem second).
>
> Method|Rel. err freq.-aware|Rel. err
> -|-|-
> PISD with Adam (proposed method)|14.70 $\pm$ 3.17 %|17.38 $\pm$ 3.75 %
> PISD with standard Adam|15.11 $\pm$ 3.83 %|21.85 $\pm$ 5.14 %
> PISD with Momentum|50.10 $\pm$ 14.45 %|42.74 $\pm$ 12.97 %
> PISD with GD|69.17 $\pm$ 19.16 %|61.11 $\pm$ 17.90 %
>
> Method|Rel. err freq.-aware|Rel. err
> -|-|-
> PISD with Adam (proposed method)|3.13 $\pm$ 1.82 %|3.82 $\pm$ 2.30 %
> PISD with standard Adam|3.64 $\pm$ 2.30 %|3.82 $\pm$ 2.49 %
> PISD with Momentum|3.08 $\pm$ 2.25 %|3.10 $\pm$ 2.31 %
> PISD with GD|4.10 $\pm$ 2.09 %|3.60 $\pm$ 1.97 %
>
> On the forward problem the optimizer variants perform comparably, but on the more challenging inverse problem the frequency-aware Adam optimizer substantially outperforms all alternatives. We will include these results in the revised version.
>
> > **Runtime results (Q5)**
>
> Runtime comparisons were limited to the Poisson and Helmholtz problems, where PISD and the baselines solve identical tasks. For Navier-Stokes, PISD generates a full 10-step trajectory whereas the baselines predict only the initial and terminal states, making a direct comparison less straightforward. We agree, however, that a runtime comparison is important also in this case and report Navier-Stokes runtimes for 500 observations in the table below, which we will include in the revised version.
>
> Method|Runtime (s)
> -|-
> PISD (full trajectory)|420
> DiffPDE (initial and terminal state)|809
> FunDPS (initial and terminal state)|246
>
> Combined with the first group of Table 4 in our paper, this shows that with less than 2 times the runtime of FunDPS, PISD achieves better accuracy on the initial and terminal states while additionally generating all intermediate timesteps.
>
> > **Sensitivity to guidance scales $\lambda_{obs}$ and $\lambda_{PDE}$ (Q6)**
>
> Thank you for this question. We report a sensitivity analysis for the forward Poisson problem with 500 observations, varying $\lambda_{obs}$ and $\lambda_{PDE}$ independently around the values used in the paper:
>
> $\lambda_{obs}$|Rel. err
> -|-
> $10^{-4}$|3.96 $\pm$ 2.21 %
> $10^{-3}$|3.15 $\pm$ 1.66 %
> $10^{-2}$|2.96 $\pm$ 1.72 %
> $10^{-1}$|3.43 $\pm$ 1.73 %
> $10^{0}$|3.48 $\pm$ 2.06 %
>
> $\lambda_{pde}$|Rel. err
> -|-
> $10^{-5}$|3.67 $\pm$ 2.24 %
> $10^{-4}$|3.40 $\pm$ 1.94 %
> $10^{-3}$|3.19 $\pm$ 1.95 %
> $10^{-2}$|2.96 $\pm$ 1.57 %
> $10^{-1}$|3.89 $\pm$ 1.89 %
>
> The results are remarkably stable across several orders of magnitude, which we attribute to the adaptive step sizes of the Adam optimizer. Due to space constraints in this rebuttal, we do not include the results for other problems but will include several in the revised version.
>
> > **Minor issues**
>
> Thank you for pointing out these issues.
> We will improve the captions of Figures 2 and 6 to clarify that the images show successive diffusion timesteps from left to right.
> We agree with the observation on the algorithm step numbering and will revise it.

---

> > ### Author Rebuttal · Reviewer_y1fm · 2026-04-05
> >
> > I would like to thank the authors for their response. Most of my major concerns are addressed. Therefore, I will increase my score and vote to accept the paper.

---

### Official Review · Reviewer_Wyvu · 2026-03-04

**Soundness:** 2
**Presentation:** 3
**Significance:** 3
**Originality:** 2
**Overall Recommendation:** 4
**Confidence:** 4

**Summary:**

The authors present Physics-Informed Spectral Diffusion (PISD), a generative framework for solving parametric partial differential equations (PDEs) under sparse observations. The method performs a diffusion process in a truncated spectral latent space (using Fourier or sine coefficients), scaling the coefficients by their empirical standard deviation to maintain controlled Sobolev regularity during the forward noise process. At inference, the model utilizes a modified Diffusion Posterior Sampling (DPS) approach, employing a frequency-aware Adam optimizer rather than standard gradient descent, to enforce both measurement conditions and physical (PDE) constraints across the entire reverse process. The method is evaluated on the Poisson, Helmholtz, and Navier-Stokes equations, demonstrating faster inference and lower reported error rates and PDE residuals compared to grid-based baselines.

**Compliance With Llm Reviewing Policy:**

Affirmed.

**Final Justification:**

The authors addressed my main concerns raised during the rebuttal. The promised ablations in the revised manuscript should further strengthen the paper.

**Key Questions For Authors:**

- Why baseline PDE residuals are very large even if they visually look fine and error rates are low? Is this discrepancy primarily due to the mismatch between exact spectral derivatives and finite-difference approximations?

- Can you train and evaluate DiffusionPDE and FunDPS on the full 10-timestep trajectories to provide a true apples-to-apples comparison of generative quality and inference speed?

- Can you provide an ablation study isolating the effect of the Adam optimizer versus the spectral representation? Specifically, how much do DiffusionPDE or FunDPS improve if they are also equipped with your frequency-aware Adam guidance?

- How much of the performance (accuracy, speed) comes from truncation level alone, per-frequency variance scaling alone, and both together?

- Could you clarify the conceptual difference and relation between your scaling approach (Lemma 3.2) and the Karhunen-Loeve expansion scaling used in previous infinite-dimensional diffusion works?

**Limitations:**

yes

**Strengths And Weaknesses:**

## **Strengths**

-  The method achieves considerable computational speed-ups, where it reduces inference time by a factor of 3 to 15 compared to baselines. It achieves this by heavily truncating the latent space (e.g. $128 \times 128$ down to $44 \times 44$ or  $32 \times 32$) while maintaining competitive or improved relative errors.

- The method transitions the paradigm to the spectral domain, directly addressing a fundamental limitation of grid-based diffusion solvers. By avoiding the spatial white noise limit inherent to grid methods, PISD naturally allows physics constraints to be applied throughout the entire reverse diffusion process.

- The paper formalizes the scaling idea via Lemma 3.2, which essentially shows that latent Gaussian noise mapped back through the scaled spectrum preserves the finite Sobolev moments of the data distribution.


## **Weaknesses**

- PISD is computing its PDE residuals naturally and exactly in the spectral domain. However, baselines (like DiffusionPDE [1], FunDPS [2]) operate on spatial grids. It is therefore unclear whether the reported improvements demonstrate true physical validity, or if they are artifact of comparing exact spectral derivatives against finite-difference approximations.

- Authors highlight that PISD generates a full 10-step temporal rollout for Navier-Stokes, whereas the baselines are restricted to initial and terminal states. However, this is not an inherent limitation of the baseline methods. One could easily train a diffusion prior on the entire trajectory by treating the 10 timesteps as channels in the data tensors. To make comparisons more apples-to-apples in NS dataset, one should train baselines on entire trajectory as well since it will also give more context information to prior.

- **Contributions are Incremental:**
    - The transition from spatial grids to spectral coefficients is a known strategy in generative modeling [3]. Its application to PDEs is highly logical given the historical success of Fourier Neural Operators [4], which reduces the novelty of PISD's fixed latent representation.
    - The rigorous theory is a good contribution. But conceptually it also mirrors other methods for dealing with infinite-dimensional Gaussian measures [5]. Scaling by the empirical variance of the data essentially maps to the Karhunen-Loeve expansion scaling by eigenvalues ($\lambda^{1/2}$) seen in prior spectral diffusion works [3].
    - Using ADAM optimizer instead of gradient updates to invoke the likelihood/measurement adherence is also not new. Applying advanced optimizers (like Adam, ADMM, L-BFGS) to explore posterior distribution has already been used in the inverse problem literature. ([6] uses ADMM; [7] uses ADAM and L-BFGS; see Algorithm 2 in [8])

- The paper lacks some of the important ablation studies to better understand where the performance gains originate. It is unclear how much of the performance gap is due to (a) the specific spectral truncation and its level (i.e. $44, 32$), (b) the variance scaling, or (c) the frequency-aware Adam optimizer as its parameters are not ablated.

- Even though traditional solvers are bad at sparse reconstruction, it would be nice to have comparisons in fully observed cases.

- **Minor Issues:**
    - Aren't $100$ or $50$ samples small for evaluations? Reported baselines used $1,000$ samples.
    - Guidance scale notations are inconsistent across main paper ($\lambda$) and some tables in appendix ($\zeta$).


[1] DiffusionPDE: Generative PDE-Solving Under Partial Observation (https://arxiv.org/abs/2406.17763)
\
[2] Guided Diffusion Sampling on Function Spaces with Applications to PDEs (https://arxiv.org/abs/2505.17004)
\
[3] Spectral Diffusion Processes (https://arxiv.org/abs/2209.14125)
\
[4] Fourier Neural Operator for Parametric Partial Differential Equations (https://arxiv.org/abs/2010.08895)
\
[5] Diffusion Generative Models in Infinite Dimensions (https://arxiv.org/abs/2212.00886)
\
[6] Solving 3D Inverse Problems using Pre-trained 2D Diffusion Models (https://arxiv.org/abs/2211.10655)
\
[7] DMPlug: A Plug-in Method for Solving Inverse Problems with Diffusion Models (https://arxiv.org/abs/2405.16749)
\
[8] Rethinking Diffusion Posterior Sampling: From Conditional Score Estimator to Maximizing a Posterior (https://arxiv.org/abs/2501.18913)

---

> ### Author Rebuttal · Authors · 2026-03-31
>
> We thank the reviewer for the detailed and constructive review.
>
> > **Comparisons of PDE residuals (Q1)**
>
> All reported PDE residuals were computed using the same finite-difference scheme for every method, including PISD (which uses spectral residuals internally during generation). There is no asymmetry in evaluation. We will clarify this in the revised version.
>
> We attribute the higher residuals of benchmark methods that work in the space-time domain to the fact that they do not generate functions of sufficient regularity: while their outputs may appear visually plausible, the underlying derivatives are poorly behaved. This is illustrated in Figure 5 in the appendix.
>
> > **Temporal rollout in Navier-Stokes (Q2)**
>
> We agree that DiffPDE and FunDPS could in principle support full trajectory generation, but neither does so for Navier-Stokes (only for the 1D Burgers equation), likely because it is difficult to stabilize or prohibitively expensive.
> A key advantage of PISD is that its compact spectral latent space makes this feasible.
> Moreover, extending the baseline architectures to 10-channel spatiotemporal fields would require non-trivial modifications, making a controlled comparison difficult.
> To nonetheless provide a more direct comparison, we refer to Q5 in our reply to Reviewer y1fm below.
>
> > **Ablation studies (Q3 \& W4)**
>
> Thank you for suggesting these important ablations.
> We report below the result of using other optimizers for FunDPS on the inverse Poisson problem with 500 observations.
>
> Method|Rel. error
> -|-
> FunDPS with GD|21.15 $\pm$ 7.31 %
> FunDPS with Momentum|18.93 $\pm$ 5.67 %
> FunDPS with ADAM|17.44 $\pm$ 4.79 %
> PISD|14.70 $\pm$ 3.17 %
>
> While Adam improves FunDPS over GD, PISD still achieves the lowest error, indicating that both the spectral representation and the optimizer contribute to the overall gains. For further optimizer ablations within PISD, see Q4 in our reply to reviewer y1fm below.
>
> > **Truncation and variance scaling (Q4)**
>
> Thank you for this question. The truncation level was chosen as a heuristic trade-off between compression and reconstruction fidelity, and we agree that a systematic study of its impact on accuracy and speed would be valuable. We will include this ablation in the revised version (it requires extensive retraining that we could not complete in time for the rebuttal).
>
> Regarding variance scaling, we have verified experimentally that removing it does not yield meaningful results, which is consistent with the theory: without variance scaling, PDE operators become ill-defined throughout the reverse process, and in the limit of infinite resolution the diffusion process itself is not well-defined.
>
> > **Relation to Karhunen-Lo\`eve expansion scaling (Q5)**
>
> Both approaches share the principle of scaling spectral basis functions. The key difference is that in prior work such as [3], the scaling factors are fixed a priori, whereas PISD learns them from the empirical data distribution. This data-driven scaling automatically adapts the generation process to the regularity of the target functions, removing a manual design choice.
>
> > **Contributions are incremental (W3)**
>
> PISD builds on existing components but combines them into a unified framework for PDE-conditioned
> diffusion. As the reviewer notes, applying spectral methods in this context is a logical direction - yet it had not been explored before this work. Our contribution is to show that data-driven spectral scaling, requiring no manual tuning, together with physics-informed guidance that is well-defined at every diffusion step, yields strong results in both accuracy and speed, with formal regularity guarantees (Lemma 3.2).
>
> We thank the reviewer for the references on advanced optimizers in diffusion-based inverse problems. While these works employ various optimizers in related settings, none of them directly replace GD with Adam within DPS as we do. We note that a concurrent independent work (AdamDPS, arXiv 2603.16797, accepted at ICLR), published after our submission, proposes exactly this replacement, which we see as validation that the idea is timely and relevant. We will add the provided references and discuss the relation in the revised version.
>
> > **Comparisons with traditional solvers (W5)**
>
> Please see our reply to an analogous question Q2 by Reviewer y1fm below.
>
> > **Number of samples for evaluations (W6.1)**
>
> A single DiffPDE evaluation takes approximately 10 minutes on our hardware, so 1000 evaluations would require around 7 days per test case. We used the same sample size across all methods for fairness. Running PISD with 1000 evaluations on the Poisson problem with 500 observations yields 3.05 $\pm$ 1.90 % (forward) and 14.48 $\pm$ 3.05 % (inverse), compared to 3.13 $\pm$ 1.82 % and 14.70 $\pm$ 3.17 % with 100, confirming that the conclusions are not significantly affected.

---

> > ### Author Rebuttal · Reviewer_Wyvu · 2026-04-04
> >
> > Thank you for the  rebuttal. I have raised my score to a 4 (Weak Accept).

---

### Official Review · Reviewer_fZtB · 2026-03-06

**Soundness:** 4
**Presentation:** 3
**Significance:** 3
**Originality:** 4
**Overall Recommendation:** 5
**Confidence:** 4

**Summary:**

This paper introduces a Physics-Informed spectral diffusion (PISD) framework, a generative latent diffusion framework for parametric partial differential equations (PDEs) conditioned on partial observations with physics-informed guidance at inference. It also shows that more advanced gradient-based optimizer such as Adam leads to significantly better results than standard gradient descent.
The author normalized each spectral coefficient in latent space, ensuring Gaussian noise in the latent space corresponds to functions with controlled regularity, and ensuring PDE operators are well defined throughout the diffusion process.
They test their method on Poisson, Helmholtz, and incompressible Navier-Stokes equation, and show the PISD method reduces inference time by a factor of 3 to 15 relative to baseline with matching or improving accuracy.

**Compliance With Llm Reviewing Policy:**

Affirmed.

**Key Questions For Authors:**

1. Do you introduce noisy or assume noiseless measurement?
2. Do you have any idea about how the method can adapt to non regular conditions or real laboratory measured spectral?
3. Could you provide a controlled breakdown about the speed comparison?

**Limitations:**

yes

**Strengths And Weaknesses:**

Strengths
The normalization ensures physics-informed guidance to be applied consistently;
It shows Adam leads to significantly better results than standard gradient descent;
It has broad evaluation across Poisson, Helmholtz, and incompressible Navier-Stokes, with both forward and inverse task;
Runs are large (50/100 independent runs), and reported speedup are large (3x vs FunDPS and 15x vs DiffusionPDE), ensuring stability and soundness

Weakness
The approach is limited to regular geometries and boundary conditions where spectral bases are available, which limits practical applications.
The speed comparison raises fairness concerns, as PISD framework has 2M parameters compared to the 54M in the DiffusionPDE and FunDPS’s network

---

> ### Author Rebuttal · Authors · 2026-03-31
>
> We thank the reviewer for their positive and constructive feedback.
>
> > **Do you introduce noisy or assume noiseless measurement?**
>
> Our framework assumes noiseless measurements. However, since related methods such as FunDPS derive similar guidance mechanisms from a Bayesian perspective that naturally accommodates noise, we expect PISD to extend to the noisy setting as well. This is an interesting idea for future research.
>
> > **Do you have any idea about how the method can adapt to non regular conditions or real laboratory measured spectral?**
>
> The restriction to regular geometries where spectral bases are available is inherent to spectral methods. Nonetheless, extensions to irregular domains are conceivable, e.g., by embedding irregular boundaries within a regular domain or by using different basis functions. Furthermore, the DPS guidance mechanism can accommodate any differentiable conditioning expressed in terms of function values or spectral coefficients, which in particular enables the incorporation of laboratory-measured spectra. These extensions are promising directions for future work.
>
> > **Fairness concerns in speed comparison / controlled breakdown.**
>
> Thank you for raising this point. The difference in model size (2M vs. 54M parameters) is not an incidental advantage but a direct consequence of our method's design: by operating in a truncated spectral latent space, PISD reduces the dimensionality of the generation problem, which in turn permits a smaller network while maintaining high accuracy. The speedup therefore reflects a genuine methodological contribution rather than an unfair implementation choice. For the baseline methods, we used the architectures and hyperparameters from their published codebases, which we consider well-optimized by their respective authors. A detailed breakdown of inference times across all methods is provided in Table 7 in the appendix.

---

> > ### Author Rebuttal · Reviewer_fZtB · 2026-04-03
> >
> > My concerns have been adequately addressed

---

### Decision · Program_Chairs · 2026-04-30

**Decision:**

Accept (regular)

**Comment:**

The average rating is 4.3 (5, 4, 4), with two reviewers raising their scores after the rebuttal and all three recommending acceptance. The main concerns (fairness of the speed comparison given the parameter-count gap (fZtB), residual computation and trajectory generation in Navier-Stokes (Wyvu), novelty relative to prior spectral diffusion and advanced-optimizer-for-DPS work (Wyvu), ablations isolating the contributions of the spectral representation vs. the optimizer (Wyvu, y1fm), and sensitivity to guidance hyperparameters (y1fm)) were adequately addressed in the rebuttal, including new experiments showing that both the spectral representation and the frequency-aware Adam contribute to the gains. Remaining limitations (restriction to regular geometries, deferred truncation-level ablation) do not undermine the core contribution: a theoretically grounded framework with consistent empirical gains and substantial speedups over state-of-the-art baselines for sparse observations. For these reasons, I recommend acceptance. I encourage the authors to implement the modifications discussed with the reviewers in the final version of the paper.